# Learning Execution through Neural Code Fusion

**Zhan Shi**[*]
The University of Texas at Austin
zshi17@utexas.edu

**Kevin Swersky, Daniel Tarlow, Parthasarathy Ranganathan, Milad Hashemi**
Google Research
{kswersky, dtarlow, parthas, miladh}@google.edu

## Abstract

As the performance of computer systems stagnates due to the end of Moore's Law, there is a need for new models that can understand and optimize the execution of general purpose code. While there is a growing body of work on using Graph Neural Networks (GNNs) to learn static representations of source code, these representations do not understand *how* code executes at runtime. In this work, we propose a new approach using GNNs to learn fused representations of general source code *and its execution*. Our approach defines a multi-task GNN over low-level representations of source code and program state (i.e., assembly code and dynamic memory states), converting complex source code constructs and data structures into a simpler, more uniform format. We show that this leads to improved performance over similar methods that do not use execution and it opens the door to applying GNN models to new tasks that would not be feasible from static code alone. As an illustration of this, we apply the new model to challenging dynamic tasks (branch prediction and prefetching) from the *SPEC CPU* benchmark suite, outperforming the state-of-the-art by 26% and 45% respectively. Moreover, we use the learned fused graph embeddings to demonstrate transfer learning with high performance on an indirectly related algorithm classification task.

## 1 Introduction

Over the last 50 years, hardware improvements have led to exponential increases in software performance, driven by Moore's Law. The end of this exponential scaling has enormous ramifications for computing (Hennessy & Patterson, 2019) since the demand for compute has simultaneously grown exponentially, relying on Moore's Law to compensate (Ranganathan, 2017). As the onus of performance optimization shifts to software, new models, representations, and methodologies for program understanding are needed to drive research and development in computer architectures, compilers, and to aid engineers in writing high performance code.

Deep learning has emerged as a powerful framework for solving difficult prediction problems across many domains, including vision (Krizhevsky et al., 2012), speech (Hinton et al., 2012), and text (Sutskever et al., 2014). Recent work has started to frame many canonical tasks in computer architecture as analogous prediction problems, and have shown that deep learning has the potential to outperform traditional heuristics (Hashemi et al., 2018). In this work, we focus on two representative tasks: address prefetching (modeling data-flow during execution) (Jouppi, 1990; Wenisch et al., 2009; Hashemi et al., 2018) and branch prediction (modeling control-flow during execution) (Jiménez & Lin, 2001; Seznec, 2011; Smith, 1981)[1]. Traditional models for solving these tasks memorize historical access patterns and branch history to make predictions about the future. However, this approach is inherently limited as there are simple cases where history-based methods cannot generalize

---

[*]Work completed during an internship at Google.

[1]As Moore's Law ends, prediction techniques in these fields have also stagnated. For example, the winner of the most recent branch prediction championship increased precision by 3.7% (Dundas, 2016).

(Section 4.6). Instead, we argue that these tasks (branch-prediction and prefetching) jointly model the intermediate behavior of a program as it executes. During execution, there is a rich and informative set of features in intermediate memory states that models can learn to drive both prediction tasks. Additionally, since programs are highly structured objects, static program syntax can supplement dynamic information with additional context about the program's execution.

We combine these two sources of information by learning a representation of a program from both its static syntax and its dynamic intermediate state during execution. This incorporates a new set of previously unexplored features for prefetching and branch prediction, and we demonstrate that these can be leveraged to obtain significant performance improvements. Inspired by recent work on learning representations of code (Allamanis et al., 2017), our approach is distinguished by two aspects. First, instead of using high level source code, we construct a new graph representation of low-level assembly code and model it with a graph neural network. Assembly makes operations like register reads, memory accesses, and branch statements explicit, naturally allowing us to model multiple problems within a single, unified representation. Second, to model intermediate state, we propose a novel snapshot mechanism that feeds limited memory states into the graph (Section 3.2).

We call our approach *neural code fusion* (NCF). This same representation can easily be leveraged for a bevy of other low-level optimizations (including: indirect branch prediction, value prediction, memory disambiguation) and opens up new possibilities for multi-task learning that were not previously possible with traditional heuristics. NCF can also be used to generate useful representations of programs for indirectly related downstream tasks, and we demonstrate this transfer learning approach on an algorithm classification problem.

On the *SPEC CPU2006* benchmarks (Sta, 2006), NCF outperforms the state-of-the-art in address and branch prediction by a significant margin. Moreover, NCF is orthogonal to existing history-based methods, and could easily combine them with our learned representations to potentially boost accuracy further. To our knowledge, NCF is the first instance of a single model that can learn simultaneously on dynamic control-flow and data-flow tasks, setting the stage for teaching neural network models to better understand how programs execute.

In summary, this paper makes the following contributions:

- An extensible graph neural network based representation of code that fuses static code and dynamic execution information into one graph.
- A binary representation for dynamic memory states that generalizes better than scalar or categorical representations.
- The first unified representation for control-flow and data-flow during program execution.
- State-of-the-art performance in branch prediction (by 26%) and prefetching (by 45%).
- We show that NCF representations pre-trained on branch prediction are useful for transfer learning, achieving competitive performance on an algorithm classification task.

## 2 BACKGROUND

In order to generate our fused representation (Figure 1), we combine three fundamental components. The representation itself builds on Graph Neural Networks (GNNs). Instead of directly representing source code, our static representation uses assembly code. To drive dynamic information through the GNN, we use binary memory snapshots. We start with background on these three components.

### 2.1 GATED GRAPH NEURAL NETWORKS

A generic graph neural network structure $G = (V, E)$ consist of a set of nodes $V$ and $K$ sets of directed edges $E = E_1, \ldots, E_K$ where $E_k \subseteq V \times V$ is the set of directed edges of type $k$. Each node $v \in V$ is annotated with a initial node embedding denoted by $x_v \in \mathbb{R}^D$ and associated with a node state vector $h_v^t \in \mathbb{R}^D$ for each step of propagation $t = 1, \ldots, T$.

Our work builds on a specific GNN variant – Gated Graph Neural Networks (GGNNs) (Li et al., 2015). GGNNs propagate information in the graph through message passing. At each step of propagation, "messages" to each node $v$ are computed as:

$$m_{kv}^t = \sum_{u:(u,v)\in E_k} f(h_u^t; \theta_k), \qquad (1)$$

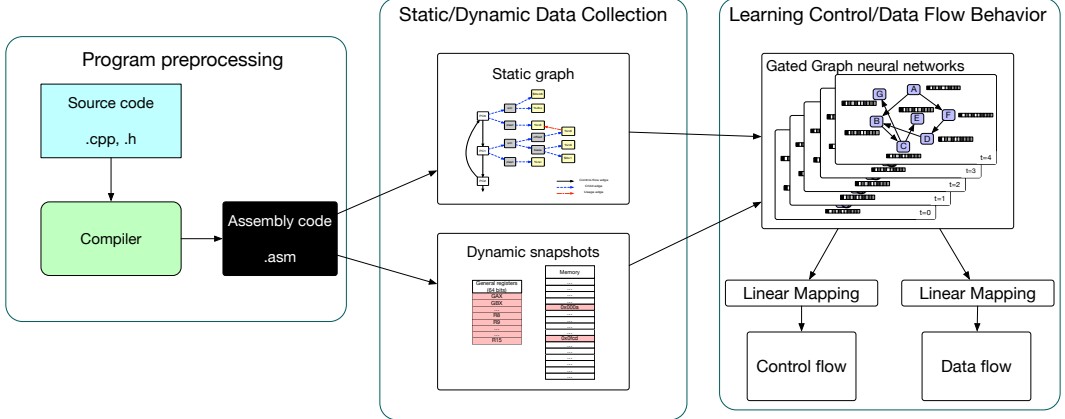

Figure 1: Overview of the fused static/dynamic graph representation.

where $m_{kv}^t$ is the zero vector if there are no edges of type $k$ directed towards $v$. $f$ is a linear layer with parameters $\theta_k$ in this model, but can be an arbitrary function. To update the state vector of a node $v$, all nonzero incoming messages are aggregated as:

$$\tilde{m}_v^t = g(\{m_{kv}^t \mid \text{for } k \text{ such that } \exists u.(u,v) \in E_k\}). \tag{2}$$

Here $g$ is an aggregation function, for which we use element-wise summation. Finally, the next state vector is computed using a gated recurrent unit (GRU) (Chung et al., 2014):

$$h_v^{t+1} = \text{GRU}(\tilde{m}_v^t, h_v^t). \tag{3}$$

The propagation is initialized with $h_v^1 = x_v$ and repeated $T$ times. The state vectors $h_v^T$ are considered as the final node embeddings. For each task, we mark a specific node $v^*$ as the "task node". We feed its final state vector $h_{v^*}^T$ to a linear output layer to make final predictions.

## 2.2 PROGRAM REPRESENTATIONS

Here we give a brief review of how compilers and processors represent source code and program state, along with tools for extracting these representations from programs and their executions.

**Dynamic Execution State.** The dynamic state of a program is the set of values that change as a program executes. This is defined by a fixed set of registers (referenced by names like `%rdi` and `%rax`) and memory (which is much larger and indexed by an integer memory address). Values are moved from memory to registers via `load` instructions and from registers to memory via `store` instructions. Finally, the *instruction pointer* specifies which instruction should be executed next.

So, what is the correct subset of dynamic state to feed into a model? In principle it could include all registers and memory. However, this can be difficult to work with (memory is very large) and it is expensive to access arbitrary memory at test time. Instead, we restrict dynamic state to a snapshot that only includes CPU general purpose registers and recently used memory states. These values are cheaply obtainable in hardware through buffers that hold recently used data and in software through dynamic instrumentation tools like Pin (see Tools section).

**Assembly Code.** Assembly code is compiled from source code and is specific to a particular processor architecture (such as x86). It is a sequence of instructions, some of which operate on register values, some of which move values between registers and memory (`loads` and `stores`), and some of which conditionally branch or jump to other locations in the program. A common way of organizing assembly code is in a *control flow graph* (CFG). Nodes of a CFG are *basic blocks*, which are sequences of instructions without any control flow statements. Edges point from a source basic block to a target basic block when it is possible for control to jump from the source bock to the target block. For x86 direct branches, there are only two possible target blocks for a given source block, which we can refer to as the *true* block and *false* block. A benefit of assembly code in our context is that it is typically less stylish and tighter to program semantics. For example, programs that are syntactically different but semantically equivalent tend to correspond to similar assembly (Figure 2).

While we only use assembly for static code, it is also possible to link assembly code to the source code it was generated from to gain additional information about high-level constructs like data structures.

| For loop | | While loop | | Assembly | | Semantics |
|---|---|---|---|---|---|---|
| for(; | i<10; i++) | while | (i<10) | 4006c1: | cmpl  $0x9,-0x4(%rbp) | # compare i and 9 (10-1) |
| { | | { | | 4006c5: | jg     4006c9 | # jump out if i >= 10 |
| | ... | | i++; | 4006cb: | addl   $0x1,-0x4(%rbp) | # add 1 to i |
| } | | } | | 4006c7: | jmp    4006c1 | # loop back |
| | | | | 4006c9: | ... | |

(a) Assembly example 1: for vs. while.

| if-else | | Ternary | | | | | | Assembly | | Semantics |
|---|---|---|---|---|---|---|---|---|---|---|
| if | (a<b) | i= | a<b | ? | a | : | b; | 4004da: | mov   -0xc(%rbp),%eax | # fetch a |
| | i = a; | | | | | | | 4004dd: | cmp   -0x8(%rbp),%eax | # compare a and b |
| else | | | | | | | | 4004e0: | jge   4004ea | # jump to i = b if a >= b |
| | i = b; | | | | | | | 4004e2: | mov   -0xc(%rbp),%eax | # fetch a |
| | | | | | | | | 4004e5: | mov   %eax,-0x4(%rbp) | # i = a |
| | | | | | | | | 4004e8: | jmp   4004f0 | # jump out |
| | | | | | | | | 4004ea: | mov   -0x8(%rbp),%eax | # fetch b |
| | | | | | | | | 4004ed: | mov   %eax,-0x4(%rbp) | # i = b |
| | | | | | | | | 4004f0: | mov   $0x1,%eax | |

(b) Assembly example 2: if-else vs. ternary.

Figure 2: Two examples where syntactically different but semantically equivalent source code is compiled to the same assembly. Corresponding sections of source/assembly are colored the same.

**Tasks.** We test learned understanding of control-flow during execution using the *branch prediction* task. Branch prediction traditionally uses heuristics to predict which target basic block will be entered next. The instruction pointer determines which basic block is currently being executed, and the target output is a boolean specifying either the true block or false block.

Branch prediction is a difficult problem with large performance implications for small relative improvements. Modern microprocessors execute hundreds of instructions speculatively, a mispredicted branch means that the processor has to discard all work completed after that branch and re-execute.

Learned understanding of data-flow during execution is tested using the *prefetching* task. Prefetching predicts the memory address that will be accessed in the next `load` operation. Since data access time is the largest bottleneck in server applications, solving data prefetching has significant implications for scaling computer architectures (Hashemi et al., 2018). Note that there is generally interleaving of branching and memory instructions, so predicting the next memory access may depend on an unknown branch decision, and vice versa.

**Tools.** Compilers convert source code into assembly code. We use *gcc*. Creating a usable snapshot of the dynamic state of a program is nontrivial. Given the large size of memory, we need to focus on memory locations that are relevant to the execution. These are obtained by monitoring the dynamic target memory addresses of `load` instructions that are executed. To obtain these snapshots, we instrument instructions during execution with a tool called Pin (Luk et al., 2005).

## 3 MODEL

We model the static assembly as a GNN (Section 3.1). Dynamic snapshots are used as features to inform the GNN of the instruction-level dynamics during execution (Section 3.2), which we show leads to model to learn the behavior of the application (Section 4).

### 3.1 GRAPH STRUCTURE

Figure 3 provides an example of our graph structure translating from 3 lines of assembly to a GNN. The graph consists of three major types of nodes: instruction nodes (in white), variable nodes (in yellow), and pseudo nodes (in grey).

*Instruction nodes* are created from instructions to serve as the backbone of the graph. Each instruction can have variable nodes or pseudo nodes as child nodes.

*Variable nodes* represent variables that use dynamic values, including registers and constants.

Instead of connecting instructions nodes directly to their child variable nodes, *Pseudo nodes* represent the sub-operations inside an instruction. The value associated with a pseudo node is computed in a bottom-up manner by recursively executing the sub-operations of its child nodes. For example, in instruction 0 in Figure 3, a pseudo node is created to represent the source operand that loads data from memory[2], which contain a child constant `0x48` and a child register `%rbx`. There are a number of different pseudo node types listed in the appendix.

Three major types of edges are used to connect nodes in the graph: control-flow edges, parent edges and usage edges. *Control-flow edges* connect an instruction node to all potential subsequent instruction nodes. For non-branch instructions, the control-flow edge from an instruction node points to the next sequential instruction node in the program. For branch instructions, control-flow edges are used to connect to both the next instruction and the branch target. *Parent edges* are used to connect child variable nodes or pseudo nodes to their parent instruction nodes or pseudo nodes. *Usage edges* provide the graph with data flow information, connecting variable nodes with their last read or write. Given this static structure, Section 3.2 describes how the GNN is initialized and used.

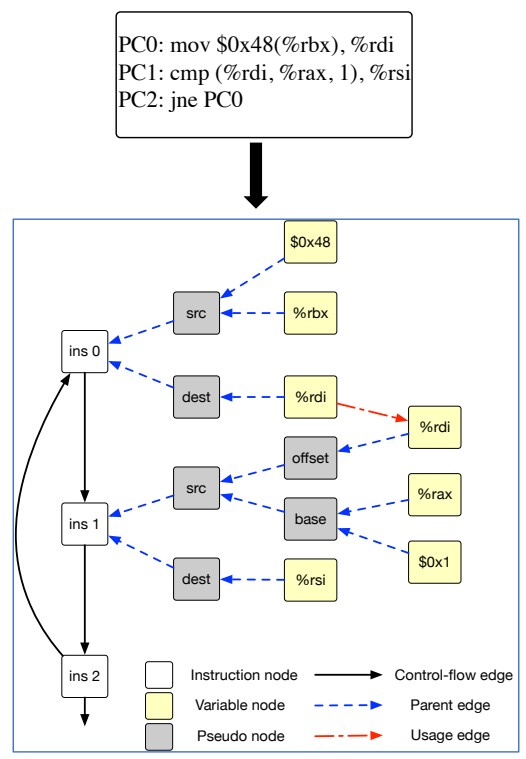

Figure 3: Graph structure on assembly code.

### 3.2 FUSED STATIC/DYNAMIC GATED GRAPH NEURAL NETWORKS

**Node initialization.** Unlike previous approaches to code analysis where node embeddings are initialized with the static text of source code, we fuse the static graph with dynamic snapshots by using dynamic state to initialize nodes in the graph.

Each variable node and pseudo node is initialized with a dynamic value from the memory snapshot. These values are converted into initial node embeddings via a learned embedding layer. We find that the numerical format of the dynamic values are critical to allowing the model to understand the application. We consider three types of representations for data values: categorical, scalar and binary. Our results (Section 4.6) show that binary has an inherent ability to generalize more efficiently than categorical or scalar representations. The intuition behind why binary generalizes so well is that the representation is inherently hierarchical, which allows for stronger generalization to previously unseen bit patterns.

Lastly, instruction nodes are initialized with zero vectors as embeddings. Given the initial embeddings, the GNN runs for a predefined number of propagation steps to obtain the final embeddings.

**Defining tasks on the graph.** Tasks are defined on nodes using masking. Similar to masking in RNNs to handle variable sequence lengths, masking in GNNs handles different numbers of task nodes. A node defined with a task has a mask value of 1 and the ones without a task are masked out using 0 during both forward and backward propagation.

Branch-prediction is defined on the branch instruction node. Since each branch can either be taken or not taken, this is a binary decision. The final node embeddings are fed into a linear layer to generate a scalar output using a sigmoid activation and a cross entropy loss.

Prefetching is defined on the *src* pseudo node that represents a memory load operation. The task is to predict the 64-bit target address of the next memory load from this node. A 64-bit output is generated by feeding the final node embeddings of the task node to a different linear layer. In this

---

[2]In x86 assembly, parentheses represent addressing memory

case, the output layer is 64-dimensional to correspond to a 64-bit address. The loss is the summation of sigmoid cross entropy loss on all 64 bits.[3]

**Scaling to large programs.** For large-scale programs, it is unrealistic to utilize the static graph built on the entire assembly file (the *gcc* benchmark has >500K instructions). As in (Li et al., 2015), to handle large graphs, only nodes which are within 100 steps to the task node will affect the prediction.

## 4 EXPERIMENTS

### 4.1 DATA COLLECTION

Our model consists of two parts, the static assembly and dynamic snapshots. To collect static assembly we use *gcc* to compile source code for each binary. This binary is then disassembled using the GNU binary utilities to obtain the assembly code.

The dynamic snapshots are captured for conditional branch and memory load instructions using the dynamic instrumentation tool Pin (Luk et al., 2005). We run the benchmarks with the reference input set and use SimPoint (Hamerly et al., 2005) to generate a single representative sample of 100 million instructions for each benchmark. Our tool attaches to the running process, fast forwards to the region of interest and outputs values of general registers and related memory addresses into a file every time the target conditional branch instructions or memory load instructions are executed by the instrumented application. We use *SPECint 2006* to evaluate our proposal. This is a standard benchmark suite commonly used to evaluate hardware and software system performance.

### 4.2 EXPERIMENTAL SETUP

We train the model on each benchmark independently. The first 70% of snapshots are used for training, and the last 30% for evaluation. Hyperparameters are reported in the appendix.

### 4.3 METRICS

To evaluate branch prediction we follow computer architecture research and use mispredictions per thousand instructions (MPKI) (Jiménez & Lin, 2001; Lee et al., 1997) as a metric. Prefetching is a harder problem as the predictor needs to accurately predict all bits of a target memory address. A prediction with even 1 bit off, especially in the high bits, is an error at often distant memory locations. We evaluate prefetching using complete accuracy, defined as an accurate prediction in all bits.

### 4.4 MODEL COMPARISONS

We compare our model to three branch predictors. The first is a bimodal predictor that uses a 2-bit saturating counter for each branch instruction to keep track of its branch history (Lee et al., 1997). The second is a widely used, state-of-the-art perceptron branch predictor (Jiménez & Lin, 2001) that uses the perceptron learning algorithm on long sequential binary taken/not-taken branch histories (Jiménez, 2016). As a more powerful baseline, we implement an offline non-linear multi-layer perceptron (MLP). The MLP has two hidden layers and each layer is of the same size as the input layer. A default SGD solver is used for optimization. The results are shown in Figure 4. We find that NCF reduces MPKI by 26% and 22% compared to the perceptron and MLP respectively. Note that some of the benchmarks (*libquantum*, *perlbench*) have zero MPKI.

Three baselines are used to evaluate our prefetching model in Figure 5. The first is a stride data prefetcher (Chen & Baer, 1995) that is good at detecting regular patterns, such as array operations. The second is a state-of-the-art address correlation (AC) prefetcher that handles irregular patterns by learning temporal address correlation (Wenisch et al., 2009). LSTM-delta is a learning-based prefetcher that captures correlation among deltas between addresses (Hashemi et al., 2018). Due to our binary representation, NCF achieves nearly 100% coverage of all addresses, unlike the 50-80% reported for the LSTM-prefetcher of (Hashemi et al., 2018). Figure 5 shows that NCF achieves significantly higher performance than prior work by handling both regular and irregular patterns with its binary representation. In both Figures 4 and 5, the applications are sorted from most-challenging to least-challenging. We find that NCF particularly outperforms the traditional baselines on the most challenging datasets. The traditional baselines in both branch prediction and prefetching leverage long sequential features. Our NCF does not yet use sequential features or sequential snapshots, we leave this for future work.

---

[3]Our framework supports multitasking in that it handles control-flow and data-flow tasks simultaneously. However, in our ablation studies, we did not see significant evidence that these tasks currently help each other.

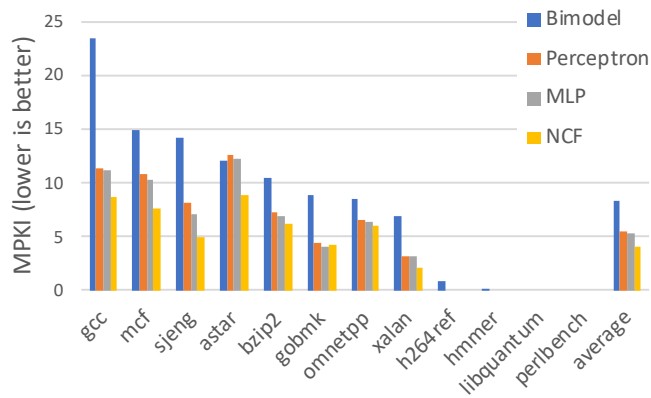

Figure 4: Evaluation of the branch-prediction task (lower is better).

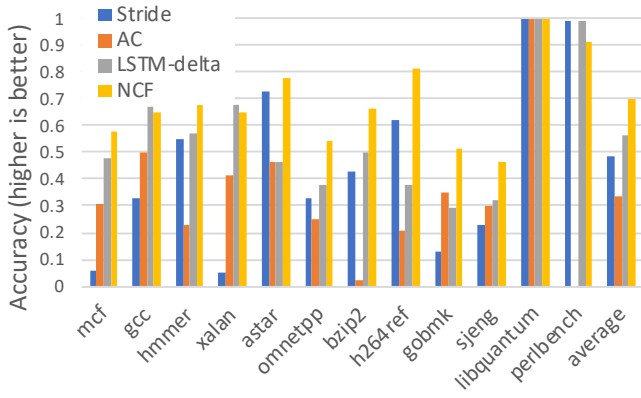

Figure 5: Evaluation of the prefetching task (higher is better).

The effectiveness of the GNN depends on the input graph, and we perform ablation studies in the appendix (Section B.1).

### 4.5 ALGORITHM CLASSIFICATION

To test if the model has learned about the behavior of the application, we test the NCF representation on an algorithm classification dataset (Lili Mou, 2016). We randomly select a subset of 15 problems from this dataset[4] and generate inputs for each program. 50 programs are randomly selected from each class. These are split into 30 for training, 10 for validation (tuning the linear SVM described below) and 10 for testing.

We generate the graph for each program post-compilation and obtain memory snapshots via our instrumentation tool. The representation is pre-trained on branch prediction and the resultant embeddings are averaged to serve as the final embedding of the program. A linear SVM is trained using the pre-trained embeddings to output a predicted class.

This yields 96.0% test accuracy, where the state-of-the-art (Ben-Nun et al., 2018) achieves 95.3% on the same subset. In contrast to Ben-Nun et al. (2018), which pre-trains an LSTM on over 50M lines of LLVM IR, our embeddings are trained on 203k lines of assembly from the algorithm classification dataset itself. This shows that branch prediction can be highly predictive of high-level program attributes, suggesting that it may be fruitful to use dynamic information to solve other static tasks.

### 4.6 GENERALIZATION TEST ON REPRESENTATIONS

Lastly, we test the effectiveness of binary representations of memory state. There are three major options for representing dynamic state: categorical, real-valued scalar, and binary. State-of-the-

---

[4]We use a subset because the programs had to be modified (by adding appropriate headers, fixing bugs) to compile and run in order to retrieve the assembly code and dynamic states.

art data prefetchers tend to use categorical representations. Recent advances in representing and manipulating numbers for neural arithmetic logic units use scalar representations (Trask et al., 2018).

We evaluate the generalization ability of these representations using a simple loop. We replace the constant 10 total iterations of the loop in Figure 2(a) with a variable $k$. The control-flow of the loop decides to stay in or jump out of the loop by comparing variable $i$ and $k$. The branch will be not taken for the first $k - 1$ times but will be taken at the $k$th time. Since traditional state-of-the-art branch predictors depend on memorizing past branch history, they will always mispredict the final branch (as it has always been taken). Our proposal is able to make the correct prediction at the $k$th time.

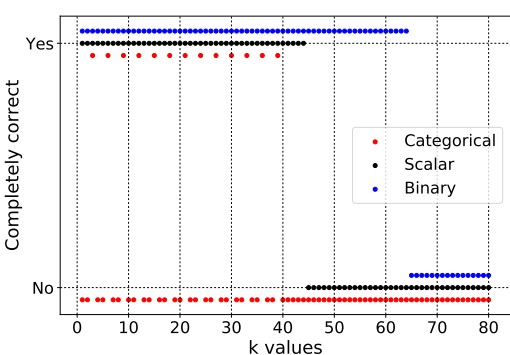

Figure 6: Generalization ability of representations. "Yes" means predictions on all branches of the loop with a $k$ value are correct.

The challenge for our model is that the value $k$ can change during program execution, and the model needs to generalize to unseen values. We use this example to test the three representations and create a testing set using $k$ values from 1 to 80. The training set only contains $k$ values from 1 to 40 with a step size of 3 (1, 4, 7, ..., 37). We feed all three representations to MLP predictors that have one hidden layer of the same size of each input representation (160 for categorical, 2 for scalar and 14 for binary). The results are shown in Figure 6.

The categorical representation can only correctly predict training samples, missing every two out of three $k$ values, where scalar and binary representations are both able to generalize across a continuous range, filling the "holes" between training samples. The binary representation generalizes to a larger range than a scalar representation, as long as the bits have been seen and toggled in the training set. Since binary is inherently hierarchical (the range increases exponentially with the number of bits), this advantage is greater in a real world 64-bit machine.

## 5 RELATED WORK

### 5.1 LEARNING FROM SOURCE CODE & EXECUTION BEHAVIOR

There is a significant body of work on learning for code, and we refer the reader to Allamanis et al. (2018) for a survey. We focus on the most relevant methods here. Li et al. (2015) use GNNs to represent the state of heap memory for a program verification application. Allamanis et al. (2017) learn to represent source code with GNNs.

Similar to us, Ben-Nun et al. (2018); Mendis et al. (2018) learn representations of code from low-level syntax, the LLVM intermediate representation (IR) or assmebly, but do not use dynamic information. We use assembly code instead of IR to maintain a 1:1 mapping between dynamic state and the static backbone of the graph (since instructions are atomic when executed). Prior work that builds graphs purely based on static source code disregard the instruction-level dynamics that are created during program execution, as a single static piece of code can execute in different ways depending on the provided inputs.

Wang et al. (2017) embed the sequences of values that variables take on during the execution of a program as a dynamic program embedding. The code is not otherwise used. The states are relatively simple (variables can take on relatively few possible values) in contrast to our dynamic states that are "from the wild." Cummins et al. (2017) embeds code and optionally allows a flat vector of auxiliary features that can depend on dynamic information. Abstract program execution can also be used as a basis for learning program representations (DeFreez et al., 2018; Henkel et al., 2018). However, neither uses concrete program state.

### 5.2 USING PROGRAM STATE TO GUIDE PROGRAM SYNTHESIS

There are several works that learn from program state to aid program synthesis (Balog et al., 2016; Parisotto et al., 2016; Devlin et al., 2017; Zohar & Wolf, 2018; Chen et al., 2019; Vijayakumar et al., 2018; Menon et al., 2013). In particular, Balog et al. (2016) use neural networks to learn a mapping from list-of-integer-valued input-output examples to the set of primitives needed. All of

these operate on programs in relatively simple Domain Specific Languages and are learning mappings from program state to code, rather than learning joint embeddings of code and program state.

## 5.3 DYNAMIC PREDICTION TASKS

Branch prediction and prefetching are heavily studied in the computer architecture domain. High-performance modern microprocessors commonly include perceptron (Jiménez & Lin, 2001) or table-based branch predictors that memorize commonly taken paths through code (Seznec, 2011).

While there has been a significant amount of work around correlation prefetching in academia (Wenisch et al., 2009; Charney & Reeves, 1995; Roth et al., 1998), modern processors only commonly implement simple stream prefetchers (Chen & Baer, 1995; Jouppi, 1990; Gindele, 1977). Recent work has related prefetching to natural language models and shown that LSTMs achieve high accuracy (Hashemi et al., 2018). However, their categorical representation covers only a limited portion of the access patterns while the binary representation described here is more general.

## 6 CONCLUSION

We develop a novel graph neural network that uses both static and dynamic features to learn a rich representation for code. Since the representation is based on a relational network, it is easy to envision extensions that include high-level source code into the model or to add new prediction tasks. Instead of focusing on hardware-realizeable systems with real-time performance, our primary focus in this paper is to develop representations that explore the limits of predictive accuracy for these problems with extremely powerful models, so that the improvements can be be eventually be distilled. This is common in machine learning research, where typically the limits of performance for a given approach are reached, and then distilled into a performant system, e.g. (Van Den Oord et al., 2016; Oord et al., 2017). However, benefits can still be derived by using the model to affect program behavior through compilation hints (Chilimbi & Hirzel, 2002; Jagannathan & Wright, 1996; Wolf et al., 1996), making this exploration immediately practical. We argue that fusing both static and dynamic features into one representation is an exciting direction to enable further progress in neural program understanding.

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

# A HYPERPARAMETERS

The hyperparameters for all models are given in Table 1.

Table 1: Hyperparameters for models.

| | | |
|---|---|---|
| | input feature size | 64 |
| | hidden size | 64 |
| Fused GGNN | propagation steps | 5 |
| | optimizer | adam |
| | learning rate | 0.01 |
| | hidden size | 2*input size |
| MLPs for generalization test | optimizer | adam |
| | L2 regularization | 0.0001 |
| SVM for algorithm classification | loss | square hinge |
| | L2 regularization | 0.01 |
| baseline: Bimodal | bits | 2 |
| | Resources | Unlimited |
| baseline: Perceptron | history length | 64 |
| | L2 regularization | 0.0001 |
| baseline: stride | Unlimited resources to store all strides (delta between addresses) for each load, predicting the most frequent stride | |
| baseline: Address Correlation | Unlimited resources to store every pairwise correlation, predicting the most frequent pair | |

## B  NODE SUB-TYPES

We describe the node sub-types in Table 2. Pseudo-nodes implement operations that are commonly known as the addressing modes of the Instruction Set Architecture. Note that node sub-types are used to derive initial node embeddings and for interpretability. They do not factor into the computation of the graph neural network.

Table 2: Descriptions about sub node types.

| Major node type | Sub-type | Description |
|---|---|---|
| Pseudo nodes | non-mem-src | a source operand that does not involve memory load operation, obtained directly from register(s) and/or constant(s) |
| | mem-src | a source operand that involves a memory load operation, obtained from loading data from a memory location |
| | non-mem-tgt | a target operand that does not involve memory write operation, writing directly to a register |
| | mem-tgt | a source operand that involves a memory write operation, writing data to a memory location |
| | base | a base that is obtained directly from a variable node |
| | ind-base | an indirect base that is obtained from certain operations on the child variable nodes, like multiplying a register by a constant |
| | offset | an offset value that is to be added to a base |
| Variable nodes | reg | a register, value is dynamically changed during execution |
| | const | a constant, value is specified in the assembly |

### B.1  ABLATION STUDY

The effectiveness of the GNN depends on the input graph. As pseudo nodes are a large component of the static graph, we run additional experiments to understand their importance. In particular, we try to only use the pseudo nodes *src* and *tgt*, which are directly connected to instruction nodes. Our data shows that removing pseudo nodes other than *src* and *tgt* and connecting variable nodes directly to *src* and *tgt* has little impact on branch prediction (an MPKI increase of 0.26), but has a large impact on the data-flow accuracy (accuracy goes down by 12.1%).

Figure 7 shows the sensitivity of task performance to the number of propagation steps during training for the GNN on *omnetpp*. We find that prefetching is more sensitive to propagation steps than branch prediction, and requires 5-8 steps for peak accuracy. Due to the control flow of programs, we find that 5-8 steps propagates information for 50-60 instruction nodes across the graph's backbone for *omnetpp* (up to 6000 nodes for *perlbench*).

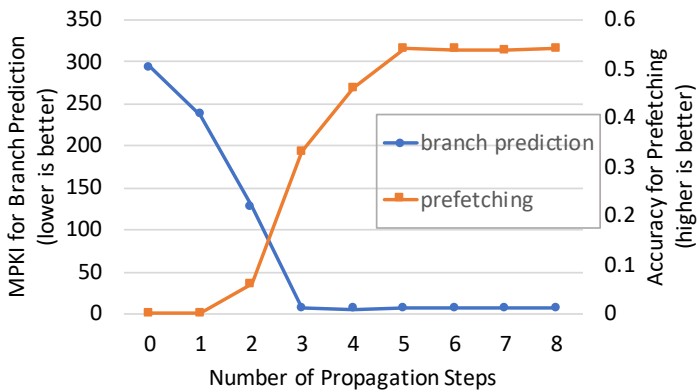

Figure 7: GNN performance vs. number of propagation steps.

