# OpenReview forum: "LEARNING EXECUTION THROUGH NEURAL CODE FUSION"
_ICLR.cc/2020/Conference — Accept (Poster)_

### Official Review · AnonReviewer3 · 2019-10-23
**Official Blind Review #3**

**Rating:** 6

**Review:**

The paper proposes using Graph Neural Networks to learn representations of source code and its execution. They test their method on the SPEC CPU benchmark suite and show substantial improvement over methods that do not use execution.

The paper's main question is to answer how to learn code representations. The main novelty introduced in their approach is to build a graph representation not of high level code but of assembly code. They also develop a way to use what they call a "snapshot mechanism" that feeds limited memory states into the graph. The downstream consequences of their methods are improved methods for example for branch prediction. Interestingly NCF can be also used to represent programs for use in downstream tasks. This is demonstrated via transfer learning in an algorithm classification problem. The paper is well written and the background / related work makes it easy for the reader to understand the problem's relevance within the related literature.

The results look well justified and empirically verified.

**Experience Assessment:**

I do not know much about this area.

**Review Assessment: Checking Correctness Of Derivations And Theory:**

N/A

**Review Assessment: Checking Correctness Of Experiments:**

I assessed the sensibility of the experiments.

**Review Assessment: Thoroughness In Paper Reading:**

N/A

---

> ### Author Response · Authors · 2019-11-08
> **Author response**
>
> Thanks for taking the time to review our work. We’d be happy to answer any questions that you may have.

---

### Official Review · AnonReviewer1 · 2019-10-24
**Official Blind Review #1**

**Rating:** 8

**Review:**

This paper presents a novel improvement in methodology for learning code execution (at the level of branch-predictions and prefetching).  They combine static program description with dynamic program state into one graph neural network, for the first time, to achieve significant performance gains on standard benchmarks.

I would vote to accept this paper.  They appear to have developed a new model structure and interface to the program information (i.e. inputs to the model), and the design decisions appear thoughtful, sensible, and well-justified (e.g. use of assembly code).  The presentation is mostly clear, with a good balance of background material, method description, and experiment results.

Taken at face value, the results are impressive, although I am not familiar enough with this field to assess the fairness of comparison against the baselines.  For example, it's a little unclear what the difference is vs previous baselines just from switching to source-code-as-input to assembly-code-as-input?

The study on memory representations (categorical vs scalar vs binary) is a helpful component which adds its own value, and the context for popularity of the alternatives is described.

Few details as to implementation are discussed, although the code is included in the submission, and after a quick glance appears substantial.

**Experience Assessment:**

I do not know much about this area.

**Review Assessment: Checking Correctness Of Derivations And Theory:**

N/A

**Review Assessment: Checking Correctness Of Experiments:**

I assessed the sensibility of the experiments.

**Review Assessment: Thoroughness In Paper Reading:**

I read the paper at least twice and used my best judgement in assessing the paper.

---

> ### Author Response · Authors · 2019-11-08
> **Author response**
>
> Thanks for taking the time to review our work, we’d be happy to answer any questions that you may have.
>
> Regarding differences vs. the other baselines, we were wondering if you could be more specific as to the baselines you’re considering. The primary baselines we compare with (for branch prediction/prefetching) do not use any source or assembly code, but rather historical decisions/access patterns. We are using a different, and potentially complementary set of features with assembly code/memory snapshots. We use assembly code because there is a clear correspondence between symbols in the code and values in the memory snapshots, but this isn't the case with source code. It isn’t clear how to get the same benefits using source code.

---

### Official Review · AnonReviewer2 · 2019-10-28
**Official Blind Review #2**

**Rating:** 3

**Review:**

Using Deep Learning and especially, GNNs seems to be a popular area of research. I am no
expert at optimizing code performance, so please take my review with a grain of salt. The algorithmic contributions of the paper are as following:

(a) GNN that combines static code and dynamic execution trace.
(b) Binary encoding of features leads to better performance in comparison to categorical and scalar representations.

The results show that the proposed method outperforms existing methods on standard benchmarks in the program execution community.

From a machine learning stand point, the contributions are straightforward and the results make sense. I have the following questions:

(I) Authors argue that binary representations are better because of their hierarchical nature. They mention that they can generalize even if not all combinations of bits are seen, but a subset is seen in a manner that every bit has been flipped a couple of times. I don’t agree with this reasoning, as seeing the individual bits flip has no guarantee that a NN would generalize to a new combination of bits unless the distance in the binary code makes sense. Is there some special way in which the binary code is constructed?

(ii) Transfer learning experiments: Its unclear to me if the comparison presented in the paper is a fair one. Comparison is made against Ben-Nun et al. pre-training on LLVM IR. I am not sure how different is LLVM IR dataset from the algorithm classification dataset. If the dataset is very different, then obviously a lot of pre-training will only result in modest performance gain. What happens with Ben-Nun method is pre-trained on the same dataset as the proposed method? Also, what is the difference in performance between the cases when the proposed method is applied to algorithm classification with and without pre-training?

Overall, the paper is a application of GNN to optimizing code execution. The technical innovations are domain-specific and do not inform the general machine learning community. Given lack of expertise in the area of program execution, I cannot judge the significance of the performance improvements reported in the paper.

Given my current concerns, I cannot recommend acceptance. I might change my ratings based on the review discussions and the author’s responses to the above questions.


**Experience Assessment:**

I do not know much about this area.

**Review Assessment: Checking Correctness Of Derivations And Theory:**

N/A

**Review Assessment: Checking Correctness Of Experiments:**

I assessed the sensibility of the experiments.

**Review Assessment: Thoroughness In Paper Reading:**

I read the paper thoroughly.

---

> ### Author Response · Authors · 2019-11-08
> **Author response**
>
> Thank you for taking the time to review the paper.
>
> Binary: We only mean this as an empirical claim. We observed surprisingly good performance with the binary representation, and we dug into how it was generalizing by running the experiments in section 4.6. These results show the generalization of the binary representation outside the training range for the k-loop example, to the highest numbers whose bits have been seen during training. We agree that there's no provable reason for why this generalization would occur, and we'll soften the claims in the paper to make it clear that this is just an empirical observation.
>
> Transfer Learning:   First, we'd like to note that our goal here is not to establish SOTA vs. Ben-Nun. Rather, the high-level idea is simply to show that it's sensible to pre-train on the execution behavior of code and then transfer the representations to a task that is typically addressed statically (just by looking at the code, not running it). This is an idea that we do not believe has previously appeared in the literature. Our goal in these experiments is just to argue that this is a potentially interesting idea going forward.
>
> Ben-Nun uses an unsupervised approach to similarly learn embeddings that can be used for indirectly related tasks via an LSTM. If we were to train both models in identical settings, it’s equivalent to removing the pre-trained embeddings (which will likely hurt the model performance), and training an LSTM on the algorithm classification dataset. We can run this experiment if the reviewer would like, but are not sure that it adds additional context, as our point is to demonstrate the richness of our graph representation.
>
> Application: We disagree that the paper is simply an application. We are tackling the research problem of how to build machine learning models that are aware of both a programmatic representation of code and its execution behavior. We study the implications on tasks related to the execution of programs (branch prediction and prefetching) and tasks that are typically done based only on the static programmatic representation of code (algorithm classification), showing that there is benefit to learning these joint representations even when the task is not explicitly about program execution. This means the results potentially have broad applicability to learning representations of programs, which is a popular and active area within the machine learning community.
>
> Intuitively, when a person wishes to understand a piece of code, they often both inspect the static source code and manually trace it with different input values. We show how to learn representations inspired by this.
>
> Finally, we believe our results demonstrating the effectiveness of the binary representation of numbers could be of interest to the broader community.
>
> The paper is related to and builds on many works in the machine learning community (please see the related work section for a more comprehensive set of references). In particular, prior work has utilized dynamic intermediate program state (Neural Program Interpreters, ICLR 2016), but their state is crafted by hand. This work is the first that leverages intermediate state from real programs (un-modified, non domain-specific languages) to build program representations. The most relevant related work was published at ICML 2018 (Hashemi et al.), NeurIPS 2018 (Ben-Nun et al, Trask et al, Chen et al), and ICLR 2018 (Allamanis et al). Many more relevant references can be found at https://ml4code.github.io/papers.html. Given the large number of publications in this area at core machine learning conferences, we believe that this paper will indeed be of interest to the general community.

---

### Decision · Program_Chairs · 2019-12-19

**Decision:**

Accept (Poster)

**Comment:**

This paper presents a method to learn representations of programs via code and execution.

The paper presents an interesting method, and results on branch prediction and address pre-fetching are conclusive. The only main critiques associated with this paper seemed to be (1) potential lack of interest to the ICLR community, and (2) lack of comparison to other methods that similarly improve performance using other varieties of information. I am satisfied by the authors' responses to these concerns, and believe the paper warrants acceptance.